# Cellulose Nitrates-Blended Composites from Bacterial and Plant-Based Celluloses

**DOI:** 10.3390/polym16091183

**Published:** 2024-04-23

**Authors:** Yulia A. Gismatulina, Vera V. Budaeva

**Affiliations:** Bioconversion Laboratory, Institute for Problems of Chemical and Energetic Technologies, Siberian Branch of the Russian Academy of Sciences (IPCET SB RAS), Biysk 659322, Russia; budaeva@ipcet.ru

**Keywords:** bacterial cellulose, oat-hull cellulose, nitration, stabilization, cellulose nitrates, nitrogen content, energetic polymers

## Abstract

Cellulose nitrates (CNs)-blended composites based on celluloses of bacterial origin (bacterial cellulose (BC)) and plant origin (oat-hull cellulose (OHC)) were synthesized in this study for the first time. Novel CNs-blended composites made of bacterial and plant-based celluloses with different BC-to-OHC mass ratios of 70/30, 50/50, and 30/70 were developed and fully characterized, and two methods were employed to nitrate the initial BC and OHC, and the three cellulose blends: the first method involved the use of sulfuric–nitric mixed acids (MAs), while the second method utilized concentrated nitric acid in the presence of methylene chloride (NA + MC). The CNs obtained using these two nitration methods were found to differ between each other, most notably, in viscosity: the samples nitrated with NA + MC had an extremely high viscosity of 927 mPa·s through to the formation of an immobile transparent acetonogel. Irrespective of the nitration method, the CN from BC (CN BC) was found to exhibit a higher nitrogen content than the CN from OHC (CN OHC), 12.20–12.32% vs. 11.58–11.60%, respectively. For the starting BC itself, all the cellulose blends of the starting celluloses and their CNs were detected using the SEM technique to have a reticulate fiber nanostructure. The cellulose samples and their CNs were detected using the IR spectroscopy to have basic functional groups. TGA/DTA analyses of the starting cellulose samples and the CNs therefrom demonstrated that the synthesized CN samples were of high purity and had high specific heats of decomposition at 6.14–7.13 kJ/g, corroborating their energy density. The CN BC is an excellent component with in-demand energetic performance; in particular, it has a higher nitrogen content while having a stable nanostructure. The CN BC was discovered to have a positive impact on the stability, structure, and energetic characteristics of the composites. The presence of CN OHC can make CNs-blended composites cheaper. These new CNs-blended composites made of bacterial and plant celluloses are much-needed in advanced, high-performance energetic materials.

## 1. Introduction

Cellulose nitrates (CNs) are known as a universal, commonly used, inflammable, hydrophobic, and plastic polymer [1] with numerous applications such as the fabrication of membranes [2], biosensors [3], paints and lacquers, filters, plastics, rocket propellants, propellants, and energetic materials [1,4,5,6,7]. The global market of CNs was estimated to be worth USD 0.86 billion in 2021 and is expected to arrive at USD 1.39 billion by 2030 [6].

The widely varying properties of CNs are determined by two main factors, the original cellulose source and the nitration method [6]. Both factors influence the structure and properties of CNs, thereby creating the diversity that leads to the wide application of CN-based materials. Cotton and wood are considered to be classical raw materials for the synthesis of CNs. Active works are currently underway on the synthesis of CNs from alternative cellulosic feedstocks such as Alfa grass fibers [8], intermediate flax [9], Miscanthus [10,11], oil palm empty fruit bunch, kenaf fibers [12], *Posidonia oceanica* [13], *Acacia mangium* [14], oat hulls [15], tobacco stalks [16], pistachio shells [17], Arabica coffee pulp [18], giant panda feces, bitter bamboo stems [19], Rhizophora, giant reed, palm leaves, esparto grass [20], and many more. The synthesis of CNs requires that cellulose-concomitant components be removed from plant raw materials, and these processes inflict environmental damage [21]. Research has focused on finding new energetic materials [22,23,24] with a high purity and a unique reticulate nanostructure [25,26] concerning nanocellulose [27,28,29] and, most notably, bacterial cellulose (BC) as the leader among the nanocelluloses [30,31,32], has presently reached its highest demand [33,34,35,36,37]. The broad application prospects and benefits of nanocellulose nitrate-based energetic materials were overviewed back in 2014 [38]. Nanocellulose nitrates have a high demand in the manufacture of ionizing radiation detectors, bioindicators, biosensors, chips, semipermeable membranes, selective sorbents, and adhesives for electronic applications [2,35,39,40].

Bacterial cellulose (BC) is the only type of nanocellulose that can be produced through biotechnology using microorganisms, resulting in hydrogels with a high purity, high mechanical strength, high molecular weight and degree of crystallinity, and an intertwined ultrafine reticulate structure [6,41,42]. The initial works on the synthesis of CN from BC (CN BC) were described as early as 2006 [22] and 2010 [23]. The next wave of larger and more extensive studies on CN BC started in 2018 [24] and continues to the present [25,26,27,28,29,30,31,32,33,34,35,36,37] because CN BC has several advantages over plant-based cellulose nitrates, in particular, its high stability due to the ultrafine, high-purity, reticulate fibrous structure.

Research on the synthesis of energetic composites is gaining relevance. For instance, previous studies [43,44] synthesized CN/nitrochitosan-blended energetic composites and demonstrated that the components have a positive interplay in the composite. It should be noted that the cellulose and chitosan were nitrated separately in those studies, and the nitrates were then blended to obtain composites through solubilization and drying. In contrast, this study prepared cellulose blends from bacterial and plant celluloses first, and the blends were then co-nitrated to furnish CNs-blended composites. The originality of this study lies in the development of new blend composites of CNs made out of bacterial and plant celluloses, making this study the first of the kind in this field. Herein, BC produced by the symbiotic microbial producer *Medusomyces gisevii* Sa-12 was employed as the cellulose of bacterial origin. Oat-hull cellulose (OHC) derived from the nitric-acid process [45] was used as the cellulose of plant origin. The choice of this plant cellulose was due to oat hulls being abundant, available, low-cost, and naturally calibrated, allowing the isolation of homogeneous fine cellulose, which implies its high capacity for a uniform distribution when preparing blended cellulose samples.

The nitration of the blended celluloses was carried out using two methods: the conventional one using sulfuric–nitric mixed acids (MAs) and the other using nitric acid in the presence of methylene chloride (NA + MC). These nitration methods were selected because the first method is available and widely employed in industry [46] and because of the ability of methylene chloride to facilitate the diffusion of the nitrating agent into cellulose in the second method. 

This study aimed to synthesize cellulose nitrates-blended composites from cellulose of bacterial and plant origin using two nitration processes. 

## 2. Materials and Methods

### 2.1. Substrates for the Study

The substrates used in this study were bacterial cellulose (BC), oat-hull cellulose (OHC), and their blended composites in BC-to-OHC ratios of 70/30, 50/50, and 30/70.

The biosynthesis of BC was performed in a synthetic nutrient medium using the symbiotic microbial producer *Medusomyces gisevii* Sa-12 using the procedure detailed in [47].

OHC was isolated using the nitric-acid process by successively treating the feedstock with dilute HNO_3_ and NaOH solutions, as described in [45]. 

#### 2.1.1. Preparing Cellulose Samples for Nitration

OHC was prepared for nitration by rubbing through a 1 mm mesh sieve. The BC gel-film was ground in a Midea MC-BL801 blender (Beijiao, China) individually (Figure 1) or with OHC added in the above-mentioned proportions until a homogeneous mass was obtained. Afterwards, the mass was poured out into silicone molds and subjected to freeze-drying in an HR7000-M freeze-drier (Harvest Right LLC, North Salt Lake, UT, USA). Figure 1 displays the appearance of the homogenized BC. The freeze-dried cellulose samples were comminuted in a LU-2605 electric mill (LUMME, Ningbo, China) to fine flakes of 1–3 mm in size. The moisture of the samples prepared for nitration was at most 5%. 

Figure 2 depicts the appearance of cellulose samples prepared for nitration: (a) plates just after freeze-drying and (b) flakes after electric milling.

#### 2.1.2. Quality Attributes of Cellulose Samples 

The quality attributes of the cellulose samples were determined using the reported procedures [48,49,50,51], as described in detail in Appendix A to this article. 

### 2.2. Nitration of Cellulose Samples

The nitration of the cellulose samples was carried out in porcelain beakers of 500 mL in volume with continuous stirring by using an HS-50A-Set vertical stirring device (Daihan (Witeg), Seoul, Republic of Korea). Two nitration methods were used: (i) sulfuric–nitric mixed acids (MA) and (ii) concentrated nitric acid in the presence of methylene chloride (NA + MC). The procedures for nitration, stabilization, and analysis of the synthesized CNs [52,53,54] are presented in Appendix A to this article.

### 2.3. Structural Study: TGA/DTA Analyses of Cellulose Samples and CNs 

After sputtering a Pt layer with a thickness of 1–5 nm, the scanning electron microscopy (SEM) analysis of celluloses and cellulose nitrates was conducted using a GSM-840 scanning electron microscope (Jeol, Tokyo, Japan) to examine the fiber surface morphology.

To investigate the molecular structure of celluloses and CNs, FTIR spectroscopy was performed using an Infralum FT-801 spectrometer (Lumex-Sibir, LLC, Novosibirsk, Russia). The spectrometer operated in the range of 4000–500 cm^−1^. For acquiring spectra, the samples were pressed into KBr tablets with a CN-to-potassium bromide ratio of 1:150.

The thermal characteristics of the cellulose samples (BC, BC/OHC = 50/50, OHC) and CN samples (CN BC, CN BC/OHC = 70/30, CN BC/OHC = 50/50, CN BC/OHC = 30/70, and CN OHC) were examined using thermogravimetric (TGA) and differential thermal (DTA) analyses. These analyses were conducted on a TGA/DTG-60 thermal analyzer (Shimadzu, Nakagyo-ku, Japan) under the following conditions: a 0.5-g sample, a 10 °C/min heating rate, a 350 °C maximum temperature, under nitrogen.

## 3. Results and Discussion

Table 1 summarizes the quality attributes of the starting cellulose samples. Only the degree of polymerization (DP) was measured for the blended celluloses.

It was experimentally found that BC exhibited high quality attributes: 99.5% α-cellulose content, 0.01% lignin content, 0.01% pentosan content, and 0.01% ash content, with the DP being 3600. OHC was inferior to BC in quality attributes but was characterized by quite high quality values for plant-based cellulose, namely, 93.8% α-cellulose content, 0.8% lignin, 0.8% pentosans, 0.31% ash, and 1450 DP. These high values of quality for OHC are comparable to those of Miscanthus cellulose [11] and higher than those of celluloses from oil palm empty fruit bunch, kenaf fibers [12], coffee pulp [18], giant panda feces, and bitter bamboo stems [19], which were successfully nitrated. The blended cellulose samples had intermediate DP values, comparable to the proportional ratio of BC to OHC.

The quality attributes of the cellulose samples aligned with the appearance, as depicted in Figure 2. For instance, the purest BC exhibiting the highest quality was the whitest cellulose. OHC containing non-cellulosic impurities looked beige-yellowish. The blended cellulose samples had a beige tint, the intensity of which increased as the percentage content of OHC in the cellulose blend rose.

The basic characteristics of the synthesized CN samples whose solubility in acetone was 100% are outlined in Table 2.

The CN samples synthesized using two methods differed between each other, first and foremost in terms of viscosity, which was extremely high during nitration using the NA + MC method and ranged between 927 mPa·s and higher, up to the formation of an acetone gel (CN BC). Due to the structuring of the CN molecules in the acetone solution, the CN acetonogel was immobile, even when the beaker was turned upside down (Appendix A), not allowing the viscosity measurement of the CN. The CN NA + MC composites also exhibited a high viscosity of 2455 mPa·s and higher. The CN-blended composite comprising CN BC 70% represented a highly viscous solution, and, similar to the acetonogel, did not allow its viscosity characteristics to be measured. Appendix A visualizes the flowability of CN solutions in acetone: the beakers were turned upside down during the viscosity measurement of the following CN NA + MC samples (from left to right): CN BC, CN BC/OHC = 70/30, CN BC/OHC = 50/50, CN BC/OHC = 30/70, and CN OHC. The acetonogel was generated, first of all, due to the high viscosity of CN NA + MC and, secondly, due to the CN BC nanoscale reticulate structure that was preserved after nitration and was predisposed to forming organogels for all the samples, except CN OHC (Appendix A). Clearly, the CN OHC sample had a viscosity below 1000 mPa·s and had no natural nanoscale reticulate structure.

The higher viscosity of CN NA + MC compared with CN MA was due to the absent hydrolytic degradation during nitration and the use of mild stabilization conditions, whereas the synthesis of CN MA involved mixed acid consisting of water and sulfuric acid and a mandatory long stabilization, which altogether reduced the viscosity of CN.

Despite CN MA exhibiting a lower viscosity compared with CN NA + MC, the CN BC MA samples were, however, characterized by a rather high viscosity of 1050 mPa·s in general, which is significantly higher than that of classical CNs [46]. The CN OHC MA sample exhibited a lower viscosity typical of CNs from plant-based cellulose [8,11,15,46]. The CNs-blended composites exhibited an intermediate viscosity between CN BC MA and CN OHC MA (200–610 mPa·s) in accordance with the proportional content of CN BC/CN OHC. Such a significant difference in viscosity between the CN samples was due to the bacterial and plant cellulose samples having different initial DPs, that is, the DP value of BC was 2.5 times that of OHC.

The increased nitrogen content observed in all CN NA + MC samples (11.60–12.32%) can be attributed to the enhanced reactivity of the initial cellulose, which was pre-soaked in methylene chloride. This pre-soaking procedure facilitates the deeper penetration of the nitrating agent into the cellulose fiber [46].

The synthesis of CN MA did not involve the organic solvent treatment step; hence, the above-described phenomenon did not occur. The interior of the fibers remained hardly accessible for the nitronium cation due to the low dielectric permeability of cellulose [55,56], leading to a lower nitrogen content of the CN samples (11.58–12.20%). That said, the difference in the nitrogen content for the CN samples from pure BC or a blend with 70% BC was more evident and was 0.12–0.20%, whereas this difference for the other CN samples was 6–10-fold smaller and was only 0.02%. This fact may suggest that the BC fibers are more amenable to the action of methylene chloride, and their reactivity increases to a greater extent than that of plant cellulose fibers. Thus, the use of the NA + MC nitration method is more advisable for the nitration of BC or, if necessary, for the synthesis of a high-viscosity acetonogel. As no information is available on this subject matter, we had no chance to compare our findings with data from the literature.

Overall, regardless of the nitration method, CN BC (12.20–12.32%) exhibited a higher nitrogen content than CN from plant cellulose (11.58–11.60%). It is expected that such a phenomenon occurs because it is widely recognized that the purity, shape, and structure of the initial cellulose significantly influence the outcomes of nitration. The high purity (Table 1) and well-developed surface of BC (Figure 3) allow one to attain higher nitrogen contents not only for CN BC, but also for the CNs-blended composites, and, consequently, improve the energetic performance of the CNs-blended composites.

As a result of the nitration process, the average molecular weight of the cellulose monomeric unit increased due to the substitution of hydrogen atoms in the cellulose hydroxyls with nitro groups. Consequently, the yield of the desired CNs experienced an increase ranging from 142% to 160% [6,46].

All the synthesized CN samples were totally acetone-soluble, evidencing the uniform nitration and the synthesis of specific CNs. 

The solubility of all the CN samples in mixed alcohol/ester was low due to the high viscosity, except for the CN from plant cellulose, CN OHC MA (91.0%), synthesized using the classical sulfuric–nitric mixed acid method. The solubility of CN in mixed alcohol/ester is well-known to depend on the nitrogen content, but this dependence is of a complex nature [46].

Figure 3 shows SEM images of the cellulose samples and their CNs.

The SEM images were taken at different zoom levels (×500–50,000) to make the difference between the CN precursors and CNs themselves more visible. The different zoom levels were able to demonstrate the fiber uniting–merging effect in the cellulose blends and their CNs. The top and bottom lines in Figure 3 show a conceptual difference in the structure and dimension of the bacterial cellulose (BC) and oat-hull cellulose (OHC) fibers: the former exhibits a reticulate fiber structure with a nanoscale fiber width, specifically 40–70 nm, while the latter consists mainly of flat cellulose fibers distinct in shape and dimensions: ribbon-shaped, spiral-shaped, or curved fibers with toothed edges, a fairly even shape, and a fiber width of 10–40 µm. Once nitrated, CN BC became structurally more compact while retaining the reticulate structure of the starting BC, in agreement with other studies [31,33,36]. When nitrated, the plant cellulose also retained the fiber shape of the starting cellulose; however, the CN fibers swelled and their surface became smoother. The fibers of CN OHC MA were more bulky and smoother compared to those of CN OHC NA + MC.

All the SEM images of the cellulose blends (Figure 3, first column) show the bacterial and plant cellulose fibers intertwined with each other. The nanoscale BC fibers intertwined with thick plant cellulose fibers. A similar fiber intertwining pattern was observed in the CNs-blended samples. It should be emphasized that the CN samples with a prevalent content of BC from 50% to 70% show a visual superiority of the bacterial fibers in interweaving with the plant fibers, whereas the CNs with a BC-to-OHC ratio of 30/70 represent thick CN OHC fibers with CN BC “nanoscale filament” fragments located chiefly on the surface of the plant fibers in the CN.

Thus, the reticulate structure of nanoscale fibers was proved to be present not only in the CN BC samples, but also in the CNs-blended samples. Therefore, there is a chance to use all these CN samples as an energetic binder gel matrix. For instance, contemporary studies [31,32,35] described a method of incorporating energetic compounds into a nanogel binder matrix based on CN BC exhibiting a three-dimensional network structure. Hence, owing to the unique network structure, the non-flowing acetonogels and high-viscosity acetonogel solutions of CN BC obtained in this study can be used as a standalone energetic gel matrix, as well as a platform for composites if combined with other energetic materials. 

Figure 4 shows the IR spectra of the starting cellulose samples and CNs therefrom.

The infrared (IR) spectra of the initial cellulose samples (Figure 4a), regardless of their origin, exhibit characteristic functional groups representative of cellulose derived from both bacterial and plant sources. These functional groups include peaks at 3339–3354 cm^−1^ (O-H stretching), 2895–2898 cm^−1^ (asymmetric and symmetric stretching of C-H), 1635–1650 cm^−1^ (O-H bending of absorbed water), 1427–1430 cm^−1^ (asymmetric bending vibration of CH_2_), 1162–1163 cm^−1^ (C-O-C stretching), and 1059–1061 cm^−1^ (skeletal stretching of C-O and vibration of β-glycosidic linkage of cellulose) [8,15,18,19,32,33,34]. Notably, the IR spectra of the cellulose samples do not exhibit stretching vibration peaks associated with impurity components, such as aromatic structures of lignin at approximately 1500 cm^−1^ and hemicelluloses at around 1700 cm^−1^. This further confirms the high quality of all the cellulose samples.

Following nitration (Figure 4b,c), the intensity of peaks corresponding to hydroxyl groups (3339–3354 cm^−1^, 2895–2898 cm^−1^) significantly decreased due to the partial substitution of hydroxyl hydrogen by the NO_2_ group in the CNs. As the degree of substitution increased, the nitrogen content of the CN samples (Table 2) also increased, while the intensity of hydroxyl peaks became less pronounced. This observation aligns with previous studies reported in [8,11,37,57,58]. Specifically, the CN NA + MC samples exhibited a lower hydroxyl intensity compared to the CN MA samples, consistent with their higher nitrogen content. On the other hand, the CN BC samples, regardless of the nitration method used, displayed the highest nitrogen content among the CN samples, resulting in the lowest peak intensity of the hydroxyl groups.

Along with that, Figure 4b,c confirm the presence of the functional groups of CN, i.e., there are two intense peaks at 1656–1659 cm^−1^ and 1278–1281 cm^−1^ attributed to the NO_2_ asymmetric and symmetric vibrations, respectively; there is a broad intense peak at 834–840 cm^−1^ attributed to the O-NO_2_ stretching vibration; and there are less intense peaks at 747–751 cm^−1^ and 683–694 cm^−1^ coming from the O-NO_2_ asymmetric and symmetric bending, respectively. These functional groups observed in the IR spectra of the CNs are consistent with those found in classical CNs derived from both bacterial sources [5,23,28,31,33,34] and plant celluloses [5,8,17,56,58].

Figure 5 displays the TGA/DTA results for three cellulose samples and CNs.

It is seen in the TGA curves shown in Figure 5a that the thermal decomposition of the starting cellulose samples occurs in three stages. The first stage demonstrates an initial small weight loss of 1–2% in the temperature region from the experiment onset to 100 °C, which was due to moisture evaporation. This is followed by a major second stage in the temperature region between 100 °C and 400 °C, where the cellulose samples decomposed with a weight loss of up to 90–92%, accompanied by endothermic transformation. The third stage ranges from 400 °C to 500 °C and the cellulose samples continue to decompose with a minor weight loss of up to 1–2%. The onset temperature of intense decomposition of the samples is 335 °C for BC, 330 °C for blended cellulose, and 323 °C for OHC. The higher decomposition temperature of cellulose evidences its purity and thermal stability [44,46,59]. Therefore, the obtained findings are consistent with the data on cellulose purity outlined in Table 1 and with the other BC study results [36,60].

The DTA curves (Figure 5d) of the cellulose samples show one endothermic decomposition peak at the following temperatures: 375 °C for BC, 371 °C for blended cellulose, and 356 °C for OHC with a sample weight loss of up to 90–92%, corroborating their purity.

The thermogravimetric analysis (TGA) of the CN samples (Figure 5b,c) revealed that the main thermal degradation of the CN samples commenced at temperatures ranging from 199–201 °C for CNs MA to 202–203 °C for CNs NA + MC. This degradation process continued up to 260–280 °C, resulting in a weight change of 70–80% for CNs MA and 80–86% for CNs NA + MC. Following this primary degradation, the CN samples exhibited a minor weight loss of 7–10% as they continued to decompose. It is worth noting that the temperature range of the decomposition onset at 199–203 °C is well-documented in the literature for both classical and alternative CN samples [8,13,15,18,19,31,58].

The differential thermal analysis (DTA) curves of the CN samples (Figure 5e,f) display a single narrow exothermic peak within the temperature range of 210–214 °C. This peak was accompanied by a decrease in the weight of the CN samples, ranging from 70 to 86%, indicating a high level of chemical purity in the synthesized CN samples. It is noteworthy that the CN BC samples obtained through both methods exhibit a narrower peak, indicating a higher purity of CN BC due to the flawless purity of BC itself (Table 1). The temperature range of the exothermic peak observed in the CN decomposition is consistent with data reported for CN derived from industrial cotton cellulose samples [5,58], other plant raw materials [8,13,15,16,19], and CN BC [23,31,36]. A comparison between the DTA curves of the CN samples and cellulose samples reveals a decrease in the decomposition temperature of CNs from 356–375 °C to 210–214 °C. This degradative behavior is attributed to the homolytic cleavage of thermally unstable O-NO_2_ bonds, which initiate autocatalytic cleavage and generate reactive radicals that accelerate the thermal decomposition of nitrated polymer chains [8,19,46,58]. The produced CN samples exhibit a high specific heat of decomposition ranging from 6.14–7.08 kJ/g for CN MA to 6.75–7.13 kJ/g for CN NA + MC, supporting their high energy density. As demonstrated by the above findings, the obtained CN samples were chemically pure and high-energy biopolymers.

As may be noted, an increase in the content of CN BC in the CNs-blended composites leads to an improvement in their thermal stability. The thermolysis of CN BC provides CN OHC with thermal resistance to decomposition, thereby slowing down the propagation of decomposition. This way, the blended CNs are more thermostable compared with CN OHC, while already having a reticulate nanostructure. This allows the properties of CNs to be improved and their application area to be expanded for use as an energetic gel matrix, basic gunpowder ingredient, selective sorbents, biosensors, semipermeable membranes, chips, bioindicators, ionizing radiation detectors, and adhesives for electronic applications [7,36,43]. CN BC is an excellent additive to achieve more promising energetic properties, in particular a higher nitrogen content with a stable nanostructure and an improved thermal stability, while the addition of CN OHC can even make the CNs-blended composite cheaper.

Based on the aforesaid findings, it can be concluded that both CN BC and CN OHC have a positive impact on the stability, structure, energetic performance, and cost of the CNs-blended composite. A similar regularity regarding the positive effect of two single components in composites was described for nitrochitosan and CN in [43,44]. 

## 4. Conclusions

This study obtained and fully characterized new composites of blended cellulose nitrates (CNs) made out of bacterial (BC) and oat-hull (OHC) celluloses with different BC-to-OHC mass ratios of 70/30, 50/50, and 30/70. The initial BC, OHC, and the three cellulose blends were nitrated using two methods: MA and NA + MC.

The CN samples synthesized using the two methods were found to differ from each other, first and foremost, in viscosity: the higher viscosity values of 927 mPa·s and above were achieved using the NA + MC nitration method, up to the formation of acetonogel (CN BC). The CN NA + MC composites exhibited a somewhat higher content of nitrogen than the CN MA: 11.60–12.32% vs. 11.58–12.20%, respectively. The CN BC was found to be richer in nitrogen content (12.20–12.32%) than the CN from plant cellulose (11.58–11.60%), irrespective of the nitration method used. This is due to the high purity and well-developed surface of BC, both individually and when blended with the other cellulose. The high nitrogen contents improve the energy performance of the CN composites.

The SEM analysis showed the cellulose samples to retain their structure upon nitration. The network fibrillar nanostructure was detected in the initial BC itself, all blended celluloses, and the CNs obtained therefrom. The IR spectroscopy showed that the CNs-blended composites synthesized herein had the representative functional groups of nitrocellulose when compared to the precursors. The TGA/DTA analyses revealed the synthesized CNs-blended composites to have a high purity, with the high specific decompositions of 6.14–7.13 kJ/g supporting their energy capacity.

The new fundamental insights gained herein towards the synthesis of CN composites from the bacterial and plant celluloses expand the knowledge of CN. The significance of the findings is the potential energetic characteristics of the new nanostructured composites of blended CNs (the role of CN BC), while simultaneously cheapening their production cost (the role of CN OHC).

There are two avenues for the future exploration of these new CNs-blended composites: civil energetic materials and biomedicine. Civil construction needs energetic materials for the precise demolition of outdated buildings where many people live. The biomedical direction is in urgent need of nitrocellulose materials for healing gels, biosensors, and wound dressings.

## Figures and Tables

**Figure 1 polymers-16-01183-f001:**
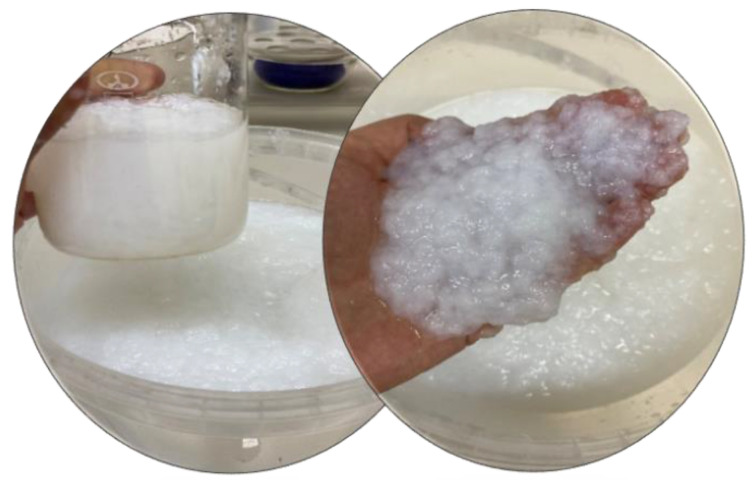
The appearance of BC ground to a homogeneous mass.

**Figure 2 polymers-16-01183-f002:**
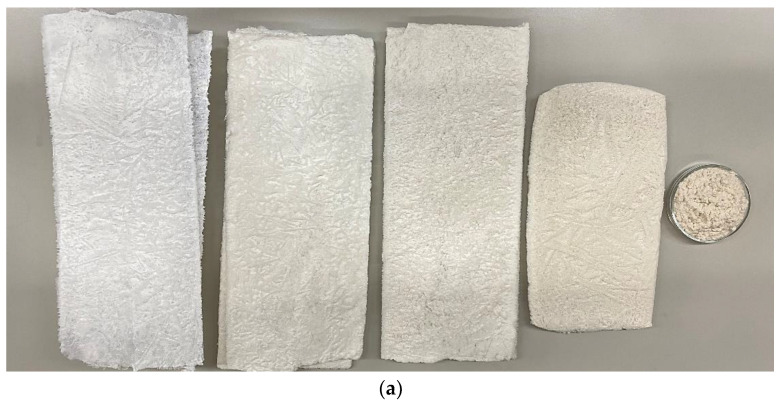
The appearance of cellulose samples prepared for nitration: (**a**) just after freeze-drying and (**b**) after electric milling (from the left to the right): BC, BC/OHC = 70/30, BC/OHC = 50/50, BC/OHC = 30/70, and OHC (rubbed through the sieve here).

**Figure 3 polymers-16-01183-f003:**
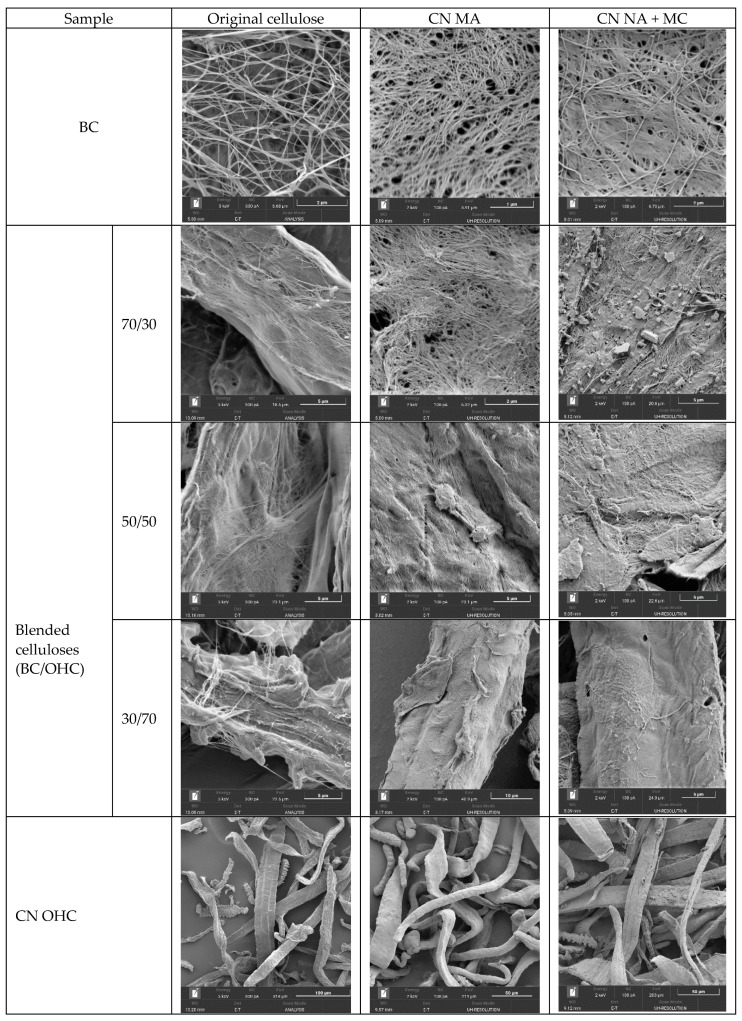
SEM images of cellulose samples and CNs therefrom.

**Figure 4 polymers-16-01183-f004:**
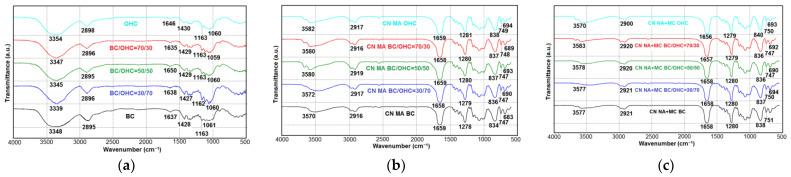
IR spectra: (**a**) starting cellulose samples, (**b**) CN MA, and (**c**) CN NA + MC.

**Figure 5 polymers-16-01183-f005:**
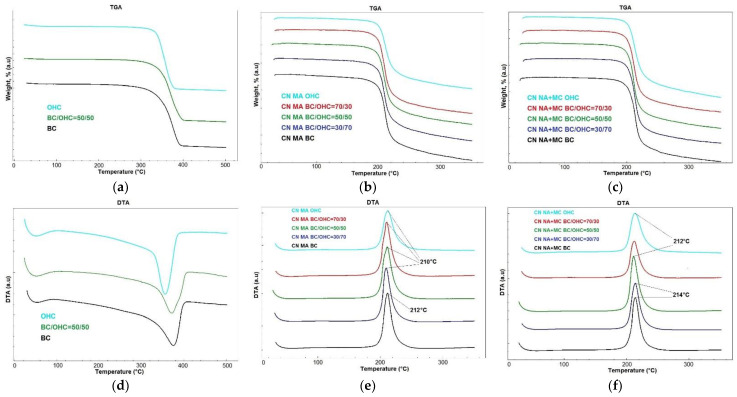
TGA and DTA images: (**a**,**d**) three cellulose samples (BC, BC/OHC = 50/50, OHC) and CN samples (CN BC, CN BC/OHC = 70/30, CN BC/OHC = 50/50, CN BC/OHC = 30/70, and CN OHC); (**b**,**e**) CN MA; and (**c**,**f**) CN NA + MC.

**Table 1 polymers-16-01183-t001:** Quality attributes of cellulose samples.

Sample		Content of Constituents *, %				DP
		α-Cellulose	Lignin	Pentosans	Ash	
BC		99.5 ± 0.1	<0.01	<0.01	<0.01	3600 ± 10
Blended cellulose (BC/OHC)	70/30	–	–	–	–	2730 ± 10
	50/50	–	–	–	–	1950 ± 10
	30/70	–	–	–	–	1540 ± 10
OHC		93.8 ± 0.1	0.8 ± 0.1	0.8 ± 0.1	0.31 ± 0.05	1450 ± 10

* On an oven-dry feedstock basis.

**Table 2 polymers-16-01183-t002:** Basic characteristics of synthesized CN samples with 100% solubility in acetone.

Sample		Nitration Method	Nitrogen Content, %	Viscosity, 2% Solution in Acetone, mPa·s	Solubility in Mixed Alcohol–Ester, %	Yield *, %
CN BC		MA	12.20 ± 0.05	1050 ± 10	11.5 ± 0.5	148 ± 2
CN from blended cellulose (BC/OHC)	70/30		11.98 ± 0.05	610 ± 5	26.3 ± 0.5	144 ± 2
	50/50		11.85 ± 0.05	530 ± 5	34.9 ± 0.5	144 ± 2
	30/70		11.74 ± 0.05	200 ± 5	50.3 ± 1.0	143 ± 2
CN OHC			11.58 ± 0.05	50 ± 5	91.0 ± 1.0	142 ± 2
CN BC		NA + MC	12.32 ± 0.05	acetonogel	0.7 ± 0.5	158 ± 2
CN from blended cellulose (BC/OHC)	70/30		12.18 ± 0.05	acetonogel	1.5 ± 0.5	160 ± 2
	50/50		11.87 ± 0.05	7450 ± 50	2.9 ± 0.5	160 ± 2
	30/70		11.76 ± 0.05	2455 ± 10	3.3 ± 0.5	159 ± 2
CN OHC			11.60 ± 0.05	927 ± 10	67.5 ± 1.0	150 ± 2

* The yield calculated after CN was open-air dried.

## Data Availability

The original contributions presented in the study are included in the article and Appendix A, further inquiries can be directed to the corresponding author.

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
