# Peer review of "Cellulose Nitrates-Blended Composites from Bacterial and Plant-Based Celluloses"

_polymers, 2024, doi:10.3390/polym16091183_

Round 1
Reviewer 1 Report
Comments and Suggestions for Authors
The researchers created cellulose nitrate composites by combining celluloses from bacterial and plant sources with different mass ratios. The nitration methods affected the viscosity of the nitrate samples, with the bacterial CN showing a higher nitrogen content. Microscopy techniques revealed a fibrillar nanostructure, while thermal analysis confirmed the high purity and energy density of the composites. These findings indicate the potential of these new materials for advanced energy applications. The study is intriguing, but it is essential to implement major adjustments and address inquiries before considering publication.
1- Page 1 line 38: avoid writing references [1-7], try to redistribute;
2- Page 2 line 55: avoid writing references [22-37], try to redistribute;
3- at the end of the introduction, you failed to mention the originality of your work in relation to previous research;
4- How do the mechanical properties of CN-blended composites based on BC and OHC compare to traditional CN-based composites or other similar materials?
5- Have you explored optimization possibilities for the synthesis processes of CNs-blended composites to enhance the uniformity and reproducibility of material properties?
6- What are the environmental implications of using bacterial and plant-origin celluloses in CN-blended composites in terms of durability and recyclability?
7- Are there specific application possibilities envisioned for these new CNs-blended composites, such as in the field of energy materials, biomedical devices, or packaging?
8- How do you plan to continue this research in the future to further explore the performance and potential applications of these CNs-blended composites?
9- Figure 6 displays the TGA/DTA results for three cellulose samples and CNs a, b and c : mention the unit of the y axis;
10- The conclusion of a scientific article aims to recap the main discoveries and outcomes of the research, analyze their importance and consequences, and offer a wider outlook on the study. It should also point out any constraints or avenues for future exploration. By doing so, the conclusion helps readers grasp the primary insights of the study and its impact on the existing knowledge in the domain.
Comments on the Quality of English LanguageThe level of quality demonstrated in the use of the English language in your text is acceptable but still requires improvements.
Author Response
Please attached.

Reviewer 2 Report
Comments and Suggestions for Authors
In this work, the authors synthesized nitrated cellulose (CN) from bacterial cellulose (BC) and oat-hull cellulose (ONC). CN blends of the two resources were characterized, showing a more promising property from CN-BC. The results are sound and comprehensive. I only have the following questions.
1. In Table 1, the lignin, pentosan, ash contents of BC were labeled as “0.01±0.1(0.05)”. However, this is physically meaningless, because one cannot have negative content percentage. Instead, the authors could have made the label of <0.01.
2. Figure 3 needs to be retaken. The quality of the image is very bad. Why use beakers with plastic bag wrappings… Please use standard vials that do not block the visions.
